# Identification of Gaussian Process State Space Models

**Stefanos Eleftheriadis**[†]**, Thomas F.W. Nicholson**[†]**, Marc P. Deisenroth**[†‡]**, James Hensman**[†]
[†]PROWLER.io,    [‡]Imperial College London
{stefanos, tom, marc, james}@prowler.io

## Abstract

The Gaussian process state space model (GPSSM) is a non-linear dynamical system, where unknown transition and/or measurement mappings are described by GPs. Most research in GPSSMs has focussed on the state estimation problem, i.e., computing a posterior of the latent state given the model. However, the key challenge in GPSSMs has not been satisfactorily addressed yet: system identification, i.e., learning the model. To address this challenge, we impose a structured Gaussian variational posterior distribution over the latent states, which is parameterised by a recognition model in the form of a bi-directional recurrent neural network. Inference with this structure allows us to recover a posterior smoothed over sequences of data. We provide a practical algorithm for efficiently computing a lower bound on the marginal likelihood using the reparameterisation trick. This further allows for the use of arbitrary kernels within the GPSSM. We demonstrate that the learnt GPSSM can efficiently generate plausible future trajectories of the identified system after only observing a small number of episodes from the true system.

## 1 Introduction

State space models can effectively address the problem of learning patterns and predicting behaviour in sequential data. Due to their modelling power they have a vast applicability in various domains of science and engineering, such as robotics, finance, neuroscience, etc. (Brown et al., 1998).

Most research and applications have focussed on linear state space models for which solutions for inference (state estimation) and learning (system identification) are well established (Kalman, 1960; Ljung, 1999). In this work, we are interested in non-linear state space models. In particular, we consider the case where a Gaussian process (GP) (Rasmussen and Williams, 2006) is responsible for modelling the underlying dynamics. This is widely known as the Gaussian process state space model (GPSSM). We choose to build upon GPs for a number of reasons. First, they are non-parametric, which makes them effective in learning from small datasets. This can be advantageous over well-known parametric models (e.g., recurrent neural networks—RNNs), especially in situation where data are not abundant. Second, we want to take advantage of the probabilistic properties of GPs. By using a GP for the latent transitions, we can get away with an approximate model and learn a distribution over functions. This allows us to account for model errors whilst quantifying uncertainty, as discussed and empirically shown by Schneider (1997) and Deisenroth et al. (2015). Consequently, the system will not become overconfident in regions of the space where data are scarce.

System identification with the GPSSM is a challenging task. This is due to un-identifiability issues: both states and transition functions are unknown. Most work so far has focused only on state estimation of the GPSSM. In this paper, we focus on addressing the challenge of system identification and based on recent work by Frigola et al. (2014) we propose a novel inference method for learning the GPSSM. We approximate the entire process of the state transition function by employing the framework of variational inference. We assume a Markov-structured Gaussian posterior distribution over the latent states. The variational posterior can be naturally combined with a recognition model

based on bi-directional recurrent neural networks, which facilitate smoothing of the state posterior over the data sequences. We present an efficient algorithm based on the reparameterisation trick for computing the lower bound on the marginal likelihood. This significantly accelerates learning of the model and allows for arbitrary kernel functions.

## 2 Gaussian process state space models

We consider the dynamical system

$$\boldsymbol{x}_t = f(\boldsymbol{x}_{t-1}, \boldsymbol{a}_{t-1}) + \boldsymbol{\epsilon}_f, \quad \boldsymbol{y}_t = g(\boldsymbol{x}_t) + \boldsymbol{\epsilon}_g, \tag{1}$$

where $t$ indexes time, $\boldsymbol{x} \in \mathbb{R}^D$ is a latent state, $\boldsymbol{a} \in \mathbb{R}^P$ are control signals (actions) and $\boldsymbol{y} \in \mathbb{R}^O$ are measurements/observations. We assume i.i.d. Gaussian system/measurement noise $\boldsymbol{\epsilon}_{(\cdot)} \sim \mathcal{N}(\boldsymbol{0}, \sigma^2_{(\cdot)} \boldsymbol{I})$. The state-space model in eq. (1) can be fully described by the measurement and transition functions, $g$ and $f$.

The key idea of a GPSSM is to model the transition function $f$ and/or the measurement function $g$ in eq. (1) using GPs, which are distributions over functions. A GP is fully specified by a mean $\eta(\cdot)$ and a covariance/kernel function $k(\cdot, \cdot)$, see e.g., (Rasmussen and Williams, 2006). The covariance function allows us to encode basic structural assumptions of the class of functions we want to model, e.g., smoothness, periodicity or stationarity. A common choice for a covariance function is the radial basis function (RBF).

Let $f(\cdot)$ denote a GP random function, and $\boldsymbol{X} = [\boldsymbol{x}_i]_{i=1}^N$ be a series of points in the domain of that function. Then, any finite subset of function evaluations, $\boldsymbol{f} = [f(\boldsymbol{x}_i)]_{i=1}^N$, are jointly Gaussian distributed

$$p(\boldsymbol{f}|\boldsymbol{X}) = \mathcal{N}(\boldsymbol{f} \,|\, \boldsymbol{\eta}, \, \boldsymbol{K}_{xx}), \tag{2}$$

where the matrix $\boldsymbol{K}_{xx}$ contains evaluations of the kernel function at all pairs of datapoints in $\mathbf{X}$, and $\boldsymbol{\eta} = [\eta(\boldsymbol{x}_i)]_{i=1}^N$ is the prior mean function. This property leads to the widely used GP regression model: if Gaussian noise is assumed, the marginal likelihood can be computed in closed form, enabling learning of the kernel parameters. By definition, the conditional distribution of a GP is another GP. If we are to observe the values $\boldsymbol{f}$ at the input locations $\boldsymbol{X}$, then we predict the values elsewhere on the GP using the conditional

$$f(\cdot) \,|\, \boldsymbol{f} \sim \mathcal{GP}\big(\eta(\cdot) + k(\cdot, \boldsymbol{X})\boldsymbol{K}_{xx}^{-1}(\boldsymbol{f} - \boldsymbol{\eta})), \, k(\cdot, \cdot) - k(\cdot, \boldsymbol{X})\boldsymbol{K}_{xx}^{-1}k(\boldsymbol{X}, \cdot)\big). \tag{3}$$

Unlike the supervised setting, in the GPSSM, we are presented with neither values of the function on which to condition, nor on *inputs* to the function since the hidden states $\boldsymbol{x}_t$ are latent. The challenge of inference in the GPSSM lies in dually inferring the latent variables $\boldsymbol{x}$ and in fitting the Gaussian process dynamics $f(\cdot)$.

In the GPSSM, we place independent GP priors on the transition function $f$ in eq. (1) for each output dimension of $\boldsymbol{x}_{t+1}$, and collect realisations of those functions in the random variables $\boldsymbol{f}$, such that

$$f_d(\cdot) \sim \mathcal{GP}\big(\eta_d(\cdot), \, k_d(\cdot, \cdot)\big), \quad \boldsymbol{f}_t = [f_d(\tilde{\boldsymbol{x}}_{t-1})]_{d=1}^D \quad \text{and} \quad p(\boldsymbol{x}_t|\boldsymbol{f}_t) = \mathcal{N}(\boldsymbol{x}_t|\boldsymbol{f}_t, \sigma_f^2 \boldsymbol{I}), \tag{4}$$

where we used the short-hand notation $\tilde{\boldsymbol{x}}_t = [\boldsymbol{x}_t, \boldsymbol{a}_t]$ to collect the state-action pair at time $t$. In this work, we use a mean function that keeps the state constant, so $\eta_d(\tilde{\boldsymbol{x}}_t) = \boldsymbol{x}_t^{(d)}$.

To reduce some of the un-identifiability problems of GPSSMs, we assume a linear measurement mapping $g$ so that the data conditional is

$$p(\boldsymbol{y}_t|\boldsymbol{x}_t) = \mathcal{N}(\boldsymbol{y}_t|\boldsymbol{W}_g \boldsymbol{x}_t + \boldsymbol{b}_g, \sigma_g^2 \boldsymbol{I}). \tag{5}$$

The linear observation model $g(\boldsymbol{x}) = \boldsymbol{W}_g \boldsymbol{x} + \boldsymbol{b}_g + \boldsymbol{\epsilon}_g$ is not limiting since a non-linear $g$ could be replaced by additional dimensions in the state space (Frigola, 2015).

### 2.1 Related work

State estimation in GPSSMs has been proposed by Ko and Fox (2009a) and Deisenroth et al. (2009) for filtering and by Deisenroth et al. (2012) and Deisenroth and Mohamed (2012) for smoothing using both deterministic (e.g., linearisation) and stochastic (e.g., particles) approximations. These

approaches focused only on inference in learnt GPSSMs and not on system identification, since learning of the state transition function $f$ without observing the system's true state $\boldsymbol{x}$ is challenging.

Towards this approach, Wang et al. (2008), Ko and Fox (2009b) and Turner et al. (2010) proposed methods for learning GPSSMs based on maximum likelihood estimation. Frigola et al. (2013) followed a Bayesian treatment to the problem and proposed an inference mechanism based on particle Markov chain Monte Carlo. Specifically, they first obtain sample trajectories from the smoothing distribution that could be used to define a predictive density via Monte Carlo integration. Then, conditioned on this trajectory they sample the model's hyper-parameters. This approach scales proportionally to the length of the time series and the number of the particles. To tackle this inefficiency, Frigola et al. (2014) suggested a hybrid inference approach combining variational inference and sequential Monte Carlo. Using the sparse variational framework from (Titsias, 2009) to approximate the GP led to a tractable distribution over the state transition function that is independent of the length of the time series.

An alternative to learning a state-space model is to follow an autoregressive strategy (as in Murray-Smith and Girard, 2001; Likar and Kocijan, 2007; Turner, 2011; Roberts et al., 2013; Kocijan, 2016), to directly model the mapping from previous to current observations. This can be problematic since noise is propagated through the system during inference. To alleviate this, Mattos et al. (2015) proposed the recurrent GP, a non-linear dynamical model that resembles a deep GP mapping from observed inputs to observed outputs, with an autoregressive structure on the intermediate latent states. They further followed the idea by Dai et al. (2015) and introduced an RNN-based recognition model to approximate the true posterior of the latent state. A downside is the requirement to feed future actions forward into the RNN during inference, in order to propagate uncertainty towards the outputs. Another issue stems from the model's inefficiency in analytically computing expectations of the kernel functions under the approximate posterior when dealing with high-dimensional latent states. Recently, Al-Shedivat et al. (2016), introduced a recurrent structure to the manifold GP (Calandra et al., 2016). They proposed to use an LSTM in order to map the observed inputs onto a non-linear manifold, where the GP actually operates on. For inefficiency, they followed an approximate inference scheme based on Kronecker products over Toeplitz-structured kernels.

## 3 Inference

Our inference scheme uses variational Bayes (see e.g., Beal, 2003; Blei et al., 2017). We first define the form of the approximation to the posterior, $q(\cdot)$. Then we derive the evidence lower bound (ELBO) with respect to which the posterior approximation is optimised in order to minimise the Kullback-Leibler divergence between the approximate and true posterior. We detail how the ELBO is estimated in a stochastic fashion and optimized using gradient-based methods, and describe how the form of the approximate posterior is given by a recurrent neural network. The graphical models of the GPSSM and our proposed approximation are shown in Figure 1.

### 3.1 Posterior approximation

Following the work by Frigola et al. (2014), we adopt a variational approximation to the posterior, assuming factorisation between the latent functions $f(\cdot)$ and the state trajectories $\boldsymbol{X}$. However, unlike Frigola et al.'s work, we do not run particle MCMC to approximate the state trajectories, but instead assume that the posterior over states is given by a Markov-structured Gaussian distribution parameterised by a recognition model (see section 3.3). In concordance with Frigola et al. (2014), we adopt a sparse variational framework to approximate the GP. The sparse approximation allows us to deal with both (a) the unobserved nature of the GP inputs and (b) any potential computational scaling issues with the GP by controlling the number of inducing points in the approximation.

The variational approximation to the GP posterior is formed as follows: Let $\boldsymbol{Z} = [\boldsymbol{z}_1, \ldots, \boldsymbol{z}_M]$ be some points in the same domain as $\tilde{\boldsymbol{x}}$. For each Gaussian process $f_d(\cdot)$, we define the inducing variables $\boldsymbol{u}_d = [f_d(\boldsymbol{z}_m)]_{m=1}^M$, so that the density of $\boldsymbol{u}_d$ under the GP prior is $\mathcal{N}(\boldsymbol{\eta}_d, \boldsymbol{K}_{zz})$, with $\boldsymbol{\eta}_d = [\eta_d(\boldsymbol{z}_m)]_{m=1}^M$. We make a mean-field variational approximation to the posterior for $\boldsymbol{U}$, taking the form $q(\boldsymbol{U}) = \prod_{d=1}^D \mathcal{N}(\boldsymbol{u}_d \,|\, \boldsymbol{\mu}_d, \boldsymbol{\Sigma}_d)$. The variational posterior of the *rest* of the points on the GP is assumed to be given by the same conditional distribution as the prior:

$$f_d(\cdot) \,|\, \boldsymbol{u}_d \sim \mathcal{GP}\big(\eta_d(\cdot) + k(\cdot, \boldsymbol{Z})\boldsymbol{K}_{zz}^{-1}(\boldsymbol{u}_d - \boldsymbol{\eta}_d), \quad k(\cdot, \cdot) - k(\cdot, \boldsymbol{Z})\boldsymbol{K}_{zz}^{-1}k(\boldsymbol{Z}, \cdot)\big). \qquad (6)$$

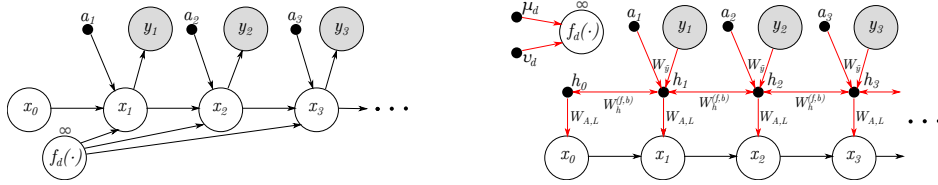

Figure 1: The GPSSM with the GP state transition functions (left), and the proposed approximation with the recognition model in the form of a bi-RNN (right). Black arrows show conditional dependencies of the model, red arrows show the data-flow in the recognition.

Integrating this expression with respect to the prior distribution $p(\boldsymbol{u}_d) = \mathcal{N}(\boldsymbol{\eta}_d, \boldsymbol{K}_{zz})$ gives the GP prior in eq. (4). Integrating with respect to the variational distribution $q(\boldsymbol{U})$ gives our approximation to the posterior process $f_d(\cdot) \sim \mathcal{GP}\big(\mu_d(\cdot), v_d(\cdot, \cdot)\big)$, with

$$\mu_d(\cdot) = \eta_d(\cdot) + k(\cdot, \boldsymbol{Z})\boldsymbol{K}_{zz}^{-1}(\boldsymbol{\mu}_d - \boldsymbol{\eta}_d), \tag{7}$$

$$v_d(\cdot, \cdot) = k(\cdot, \cdot) - k(\cdot, \boldsymbol{Z})\boldsymbol{K}_{zz}^{-1}[\boldsymbol{K}_{zz} - \boldsymbol{\Sigma}_d]\boldsymbol{K}_{zz}^{-1}k(\boldsymbol{Z}, \cdot). \tag{8}$$

The approximation to the posterior of the state trajectory is assumed to have a Gauss-Markov structure:

$$q(\boldsymbol{x}_0) = \mathcal{N}\big(\boldsymbol{x}_0 \,|\, \boldsymbol{m}_0, \boldsymbol{L}_0\boldsymbol{L}_0^\top\big), \quad q(\boldsymbol{x}_t \,|\, \boldsymbol{x}_{t-1}) = \mathcal{N}\big(\boldsymbol{x}_t \,|\, \boldsymbol{A}_t\boldsymbol{x}_{t-1}, \boldsymbol{L}_t\boldsymbol{L}_t^\top\big). \tag{9}$$

This distribution is specified through a single mean vector $\boldsymbol{m}_0$, a series of square matrices $\boldsymbol{A}_t$, and a series of lower-triangular matrices $\boldsymbol{L}_t$. It serves as a locally linear approximation to an overall non-linear posterior over the states. This is a good approximation provided that the $\Delta t$ between the transitions is sufficiently small.

With the approximating distributions for the variational posterior defined in eq. (7)–(9), we are ready to derive the evidence lower bound (ELBO) on the model's true likelihood. Following (Frigola, 2015, eq. (5.10)), the ELBO is given by

$$\begin{aligned} \text{ELBO} = \; &\mathbb{E}_{q(\boldsymbol{x}_0)}[\log p(\boldsymbol{x}_0)] + \text{H}[q(\boldsymbol{X})] - \text{KL}[q(\boldsymbol{U}) \,||\, p(\boldsymbol{U})] \\ &+ \mathbb{E}_{q(\boldsymbol{X})}\Big[ \sum_{t=1}^{T}\sum_{d=1}^{D} -\frac{1}{2\sigma_f^2}v_d(\tilde{\boldsymbol{x}}_{t-1}, \tilde{\boldsymbol{x}}_{t-1}) + \log\mathcal{N}\big(x_t^{(d)} \,|\, \mu_d(\tilde{\boldsymbol{x}}_{t-1}), \sigma_f^2\big)\Big] \\ &+ \mathbb{E}_{q(\boldsymbol{X})}\Big[ \sum_{t=1}^{T} \log\mathcal{N}\big(\boldsymbol{y}_t \,|\, g(\boldsymbol{x}_t), \sigma_g^2\boldsymbol{I}_O\big)\Big], \end{aligned} \tag{10}$$

where $\text{KL}[\cdot\,||\,\cdot]$ is the Kullback-Leibler divergence, and $\text{H}[\cdot]$ denotes the entropy. Note that with the above formulation we can naturally deal with multiple episodic data since the ELBO can be factorised across independent episodes. We can now learn the GPSSM by optimising the ELBO w.r.t. the parameters of the model and the variational parameters. A full derivation is provided in the supplementary material.

The form of the ELBO justifies the Markov-structure that we have assumed for the variational distribution $q(\boldsymbol{X})$: we see that the latent states only interact over pairwise time steps $\boldsymbol{x}_t$ and $\boldsymbol{x}_{t-1}$; adding further structure to $q(\boldsymbol{X})$ is unnecessary.

## 3.2 Efficient computation of the ELBO

To compute the ELBO in eq. (10), we need to compute expectations w.r.t. $q(\boldsymbol{X})$. Frigola et al. (2014) showed that for the RBF kernel the relevant expectations can be computed in closed form in a similar way to Titsias and Lawrence (2010). To allow for general kernels we propose to use the reparameterisation trick (Kingma and Welling, 2014; Rezende et al., 2014) instead: by sampling a single trajectory from $q(\boldsymbol{X})$ and evaluating the integrands in eq. (10), we obtain an unbiased estimate of the ELBO. To draw a sample from the Gauss-Markov structure in eq. (9), we first sample $\boldsymbol{\epsilon}_t \sim \mathcal{N}(\boldsymbol{0}, \boldsymbol{I})$, $t = 0, \ldots, T$, and then apply recursively the affine transformation

$$\boldsymbol{x}_0 = \boldsymbol{m}_0 + \boldsymbol{L}_0\boldsymbol{\epsilon}_0, \quad \boldsymbol{x}_t = \boldsymbol{A}_t\boldsymbol{x}_{t-1} + \boldsymbol{L}_t\boldsymbol{\epsilon}_t. \tag{11}$$

This simple estimator of the ELBO can then be used in optimisation using stochastic gradient methods; we used the Adam optimizer (Kingma and Ba, 2015). It may seem initially counter-intuitive to use a stochastic estimate of the ELBO where one is available in closed form, but this approach offers two distinct advantages. First, computation is dramatically reduced: our scheme requires $\mathcal{O}(TD)$ storage in order to evaluate the integrand in eq. (10) at a single sample from $q(\boldsymbol{X})$. A scheme that computes the integral in closed form requires $\mathcal{O}(TM^2)$ (where M is the number of inducing variables in the sparse GP) storage for the sufficient statistics of the kernel evaluations. The second advantage is that we are no longer restricted to the RBF kernel, but can use any valid kernel for inference and learning in GPSSMs. The reparameterisation trick also allows us to perform batched updates of the model parameters, amounting to doubly stochastic variational inference (Titsias and Lázaro-Gredilla, 2014), which we experimentally found to improve run-time and sample-efficiency.

Some of the elements of the ELBO in eq. (10) are still available in closed-form. To reduce the variance of the estimate of the ELBO we exploit this where possible: the entropy of the Gauss-Markov structure is $\mathrm{H}[q(\boldsymbol{X})] = -\frac{TD}{2} \log(2\pi e) - \sum_{t=0}^{T} \log(\det(\boldsymbol{L}_t))$; the expected likelihood (last term in eq. (10)) can be computed easily given the marginals of $q(\boldsymbol{X})$, which are given by

$$q(\boldsymbol{x}_t) = \mathcal{N}(\boldsymbol{m}_t, \boldsymbol{\Sigma}_t), \quad \boldsymbol{m}_t = \boldsymbol{A}_t \boldsymbol{m}_{t-1}, \quad \boldsymbol{\Sigma}_t = \boldsymbol{A}_t \boldsymbol{\Sigma}_{t-1} \boldsymbol{A}_t^\top + \boldsymbol{L}_t \boldsymbol{L}_t^\top , \tag{12}$$

and the necessary Kullback-Leibler divergences can be computed analytically: we use the implementations from GPflow (Matthews et al., 2017).

### 3.3 A recurrent recognition model

The variational distribution of the latent trajectories in eq. (9) has a large number of parameters $(\boldsymbol{A}_t, \boldsymbol{L}_t)$ that grows with the length of the dataset. Further, if we wish to train a model on multiple episodes (independent data sequences sharing the same dynamics), then the number of parameters grows further. To alleviate this, we propose to use a recognition model in the form of a bi-directional recurrent neural network (bi-RNN), which is responsible for recovering the variational parameters $\boldsymbol{A}_t, \boldsymbol{L}_t$.

A bi-RNN is a combination of two independent RNNs operating on opposite directions of the sequence. Each network is specified by two weight matrices $\boldsymbol{W}$ acting on a hidden state $\boldsymbol{h}$:

$$\boldsymbol{h}_t^{(f)} = \phi(\boldsymbol{W}_h^{(f)} \boldsymbol{h}_{t-1}^{(f)} + \boldsymbol{W}_{\tilde{y}}^{(f)} \tilde{\boldsymbol{y}}_t + \boldsymbol{b}_h^{(f)}), \quad \text{forward passing} \tag{13}$$

$$\boldsymbol{h}_t^{(b)} = \phi(\boldsymbol{W}_h^{(b)} \boldsymbol{h}_{t+1}^{(b)} + \boldsymbol{W}_{\tilde{y}}^{(b)} \tilde{\boldsymbol{y}}_t + \boldsymbol{b}_h^{(b)}), \quad \text{backward passing} \tag{14}$$

where $\tilde{\boldsymbol{y}}_t = [\boldsymbol{y}_t, \boldsymbol{a}_t]$ denotes the concatenation of the observed data and control actions and the superscripts denote the direction (forward/backward) of the RNN. The activation function $\phi$ (we use the $\tanh$ function), acts on each element of its argument separately. In our experiments we found that using gated recurrent units (Cho et al., 2014) improved performance of our model. We now make the parameters of the Gauss-Markov structure dependent on the sequences $\boldsymbol{h}^{(f)}, \boldsymbol{h}^{(b)}$, so that

$$\boldsymbol{A}_t = \text{reshape}(\boldsymbol{W}_A[\boldsymbol{h}_t^{(f)}; \boldsymbol{h}_t^{(b)}] + \boldsymbol{b}_A), \quad \boldsymbol{L}_t = \text{reshape}(\boldsymbol{W}_L[\boldsymbol{h}_t^{(f)}; \boldsymbol{h}_t^{(b)}] + \boldsymbol{b}_L) . \tag{15}$$

The parameters of the Gauss-Markov structure $q(\boldsymbol{X})$ are now almost completely encapsulated in the recurrent recognition model as $\boldsymbol{W}_h^{(f,b)}, \boldsymbol{W}_{\tilde{y}}^{(f,b)}, \boldsymbol{W}_A, \boldsymbol{W}_L, \boldsymbol{b}_h^{(f,b)}, \boldsymbol{b}_A, \boldsymbol{b}_L$. We only need to infer the parameters of the initial state, $\boldsymbol{m}_0, \boldsymbol{L}_0$ for each episode; this is where we utilise the functionality of the bi-RNN structure. Instead of directly learning the initial state $q(\boldsymbol{x}_0)$, we can now obtain it indirectly via the output state of the backward RNN. Another nice property of the proposed recognition model is that now $q(\boldsymbol{X})$ is recognised from both future and past observations, since the proposed bi-RNN recognition model can be regarded as a forward and backward sequential smoother of our variational posterior. Finally, it is worth noting the interplay between the variational distribution $q(\boldsymbol{X})$ and the recognition model. Recall that the variational distribution is a Bayesian linear approximation to the non-linear posterior and is fully defined by the time varying parameters, $\boldsymbol{A}_t, \boldsymbol{L}_t$; the recognition model has the role to recover these parameters via the non-linear and time invariant RNN.

## 4 Experiments

We benchmark the proposed GPSSM approach on data from one illustrative example and three challenging non-linear data sets of simulated and real data. Our aim is to demonstrate that we can: (i)

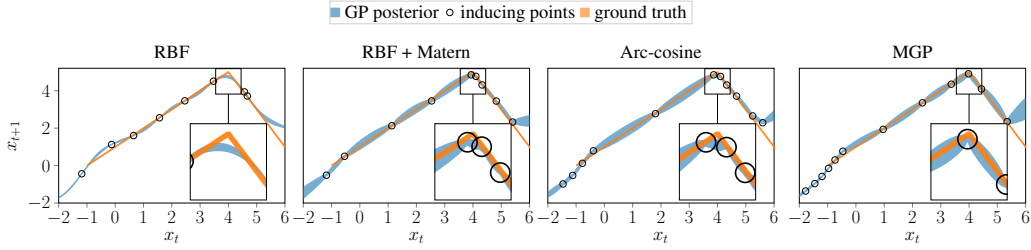

Figure 2: The learnt state transition function with different kernels. The true function is given by eq. (16).

benefit from the use of non-smooth kernels with our approximate inference and accurately model non-smooth transition functions; (ii) successfully learn non-linear dynamical systems even from noisy and partially observed inputs; (iii) sample plausible future trajectories from the system even when trained with either a small number of episodes or long time sequences.

### 4.1 Non-linear system identification

We first apply our approach to a synthetic dataset generated broadly according to (Frigola et al., 2014). The data is created using a non-linear, non-smooth transition function with additive state and observation noise according to: $p(x_{t+1}|x_t) = \mathcal{N}(f(x_t), \sigma_f^2)$, and $p(y_t|x_t) = \mathcal{N}(x_t, \sigma_g^2)$, where

$$f(x_t) = x_t + 1, \quad \text{if } x_t < 4, \qquad 13 - 2x_t, \quad \text{otherwise} . \tag{16}$$

In our experiments, we set the system and measurement noise variances to $\sigma_f^2 = 0.01$ and $\sigma_g^2 = 0.1$, respectively, and generate 200 episodes of length 10 that were used as the observed data for training the GPSSM. We used 20 inducing points (initialised uniformly across the range of the input data) for approximating the GP and 20 hidden units for the recurrent recognition model. We evaluate the following kernels: RBF, additive composition of the RBF (initial $\ell = 10$) and Matern ($\nu = \frac{1}{2}$, initial $\ell = 0.1$), 0-order arc-cosine (Cho and Saul, 2009), and the MGP kernel (Calandra et al., 2016) (depth 5, hidden dimensions $[3, 2, 3, 2, 3]$, $\tanh$ activation, Matern ($\nu = \frac{1}{2}$) compound kernel).

The learnt GP state transition functions are shown in Figure 2. With the non-smooth kernels we are able to learn accurate transitions and model the instantaneous dynamical change, as opposed to the smooth transition learnt with the RBF. Note that all non-smooth kernels place inducing points directly on the peak (at $x_t = 4$) to model the kink, whereas the RBF kernel explains this behaviour as a longer-scale wiggliness of the posterior process. When using a kernel without the RBF component the GP posterior quickly reverts to the mean function ($\eta(x) = x$) as we move away from the data: the short length-scales that enable them to model the instantaneous change prevent them from extrapolating downwards in the transition function. The composition of the RBF and Matern kernel benefits from long and short length scales and can better extrapolate. The posteriors can be viewed across a longer range of the function space in the supplementary material.

### 4.2 Modelling cart-pole dynamics

We demonstrate the efficacy of the proposed GPSSM on learning the non-linear dynamics of the cart-pole system from (Deisenroth and Rasmussen, 2011). The system is composed of a cart running on a track, with a freely swinging pendulum attached to it. The state of the system consists of the cart's position and velocity, and the pendulum's angle and angular velocity, while a horizontal force (action) $a \in [-10, 10]N$ can be applied to the cart. We used the PILCO algorithm from (Deisenroth and Rasmussen, 2011) to learn a feedback controller that swings the pendulum and balances it in the inverted position in the middle of the track. We collected trajectory data from 16 trials during learning; each trajectory/episode was $4\,\mathrm{s}$ (40 time steps) long.

When training the GPSSM for the cart-pole system we used data up to the first 15 episodes. We used 100 inducing points to approximate the GP function with a Matern $\nu = \frac{1}{2}$ and 50 hidden units for the recurrent recognition model. The learning rate for the Adam optimiser was set to $10^{-3}$. We qualitatively assess the performance of our model by feeding the control sequence of the last episode to the GPSSM in order to generate future responses.

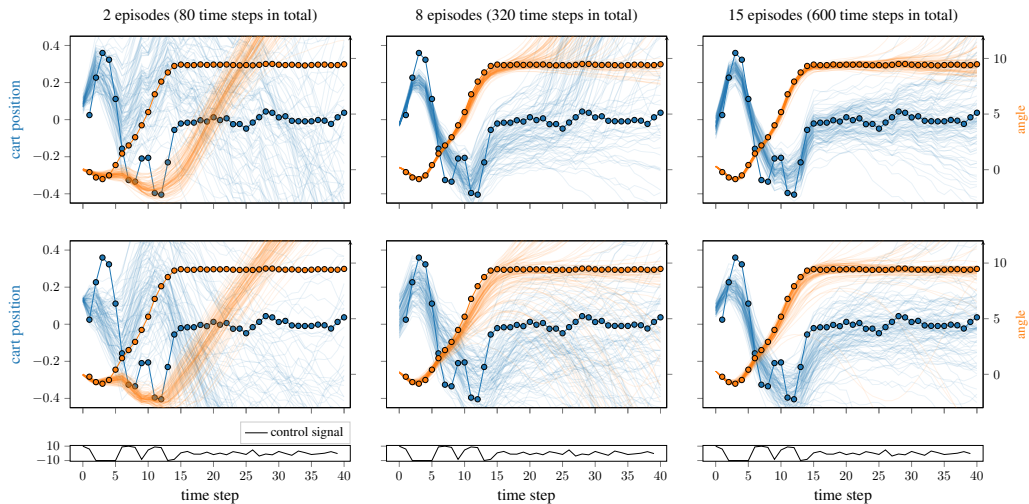

Figure 3: Predicting the cart's position and pendulum's angle behaviour from the cart-pole dataset by applying the control signal of the testing episode to sampled future trajectories from the proposed GPSSM. Learning of the dynamics is demonstrated with *observed* (upper row) and *hidden* (lower row) velocities and with increasing number of training episodes. Ground truth is denoted with the marked lines.

In Figure 3, we demonstrate the ability of the proposed GPSSM to learn the underlying dynamics of the system from a different number of episodes with fully and partially observed data. In the top row, the GPSSM observes the full 4D state, while in the bottom row, we train the GPSSM with only the cart's position and the pendulum's angle observed (i.e., the true state is not fully observed since the velocities are hidden). In both cases, sampling long-term trajectories based on only 2 episodes for training does not result in plausible future trajectories. However, we could model part of the dynamics after training with only 8 episodes (320 time steps interaction with the system), while training with 15 episodes (600 time steps in total) allowed the GPSSM to produce trajectories similar to the ground truth. It is worth emphasising the fact that the GPSSM could recover the unobserved velocities in the latent states, which resulted in smooth transitions of the cart and swinging of the pendulum. However, it seems that the recovered cart's velocity is overestimated. This is evidenced by the increased variance in the prediction of the cart's position around 0 (the centre of the track). Detailed fittings for each episode and learnt latent states with observed and hidden velocities are provided in the supplementary material.

Table 1: Average Euclidean distance between the true and the predicted trajectories, measured at the pendulum's tip. The error is in pendulum's length units.

| | 2 episodes | 8 episodes | 15 episodes |
|---|---|---|---|
| Kalman | 1.65 | 1.52 | 1.48 |
| ARGP | **1.22** | 1.03 | 0.80 |
| GPSSM | 1.21 | **0.67** | **0.59** |

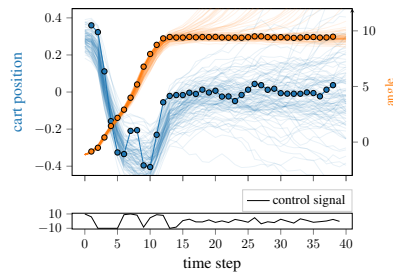

Figure 4: Predictions with lagged actions.

In Table 1, we provide the average Euclidean distance between the predicted and the true trajectories measured at the pendulum's tip, with fully observed states. We compare to two baselines: (i) the auto-regressive GP (ARGP) that maps the tuple $[\boldsymbol{y}_{t-1}, a_{t-1}]$ to the next observation $\boldsymbol{y}_t$ (as in PILCO (Deisenroth et al., 2015)), and (ii) a linear system for identification that uses the Kalman filtering technique (Kalman, 1960). We see that the GPSSM significantly outperforms the baselines on this highly non-linear benchmark. The linear system cannot learn the dynamics at all, while the ARGP only manages to produce sensible error (less than a pendulum's length) after seeing 15 episodes. Note

that the GPSSM trained on 8 episodes produces trajectories with less error than the ARGP trained on 15 episodes.

We also ran experiments using lagged actions where the partially observed state at time $t$ is affected by the action at $t - 2$. Figure 4 shows that we are able to sample future trajectories with an accuracy similar to time-aligned actions. This indicates that our model is able to learn a compressed representation of the full state and previous inputs, essentially 'remembering' the lagged actions.

### 4.3 Modelling double pendulum dynamics

We demonstrate the learning and modelling of the dynamics of the double pendulum system from (Deisenroth et al., 2015). The double pendulum is a two-link robot arm with two actuators. The state of the system consists of the angles and the corresponding angular velocities of the inner and outer link, respectively, while different torques $a_1, a_2 \in [-2, 2]$ Nm can be applied to the two actuators. The task of swinging the double pendulum and balancing it in the upwards position is extremely challenging. First, it requires the interplay of two correlated control signals (i.e., the torques). Second, the behaviour of the system, when operating at free will, is chaotic.

We learn the underlying dynamics from episodic data (15 episodes, 30 time steps long each). Training of the GPSSM was performed with data up to 14 episodes, while always demonstrating the learnt underlying dynamics on the last episode, which serves as the test set. We used 200 inducing points to approximate the GP function with a Matern $\nu = \frac{1}{2}$ and 80 hidden units for the recurrent recognition model. The learning rate for the Adam optimiser was set to $10^{-3}$. The difficulty of the task is evident in Figure 5, where we can see that even after observing 14 episodes we cannot accurately predict the system's future behaviour for more than 15 time steps (i.e., $1.5$ s). It is worth noting that we can generate reliable simulation even though we observe only the pendulums' angles.

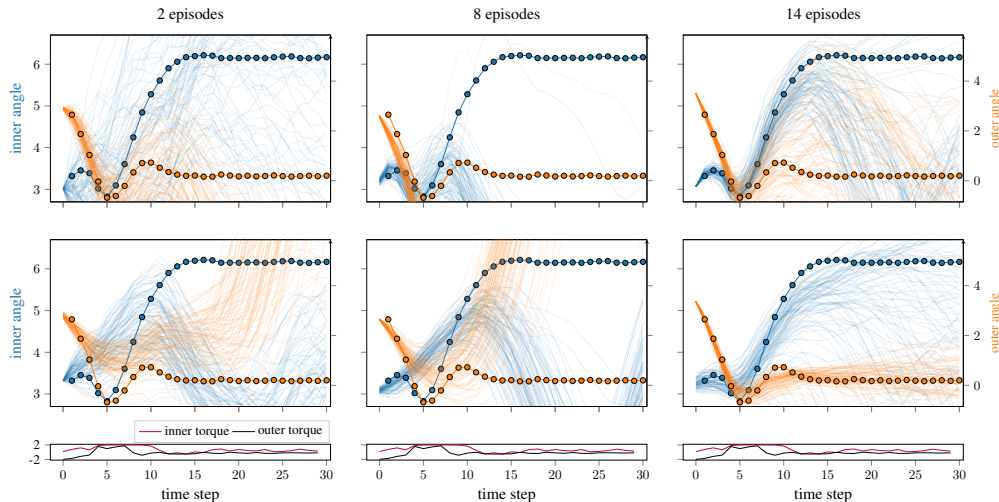

Figure 5: Predicting the inner and outer pendulum's angle from the double pendulum dataset by applying the control signals of the testing episode to sampled future trajectories from the proposed GPSSM. Learning of the dynamics is demonstrated with *observed* (upper row) and *hidden* (lower row) angular velocities and with increasing number of training episodes. Ground truth is denoted with the marked lines.

### 4.4 Modelling actuator dynamics

Here we evaluate the proposed GPSSM on real data from a hydraulic actuator that controls a robot arm (Sjöberg et al., 1995). The input is the size of the actuator's valve opening and the output is its oil pressure. We train the GPSSM on half the sequence (512 steps) and evaluate the model on the remaining half. We use 15 inducing points to approximate the GP function with a combination of an RBF and a Matern $\nu = \frac{1}{2}$ and 15 hidden units for the recurrent recognition model. Figure 6

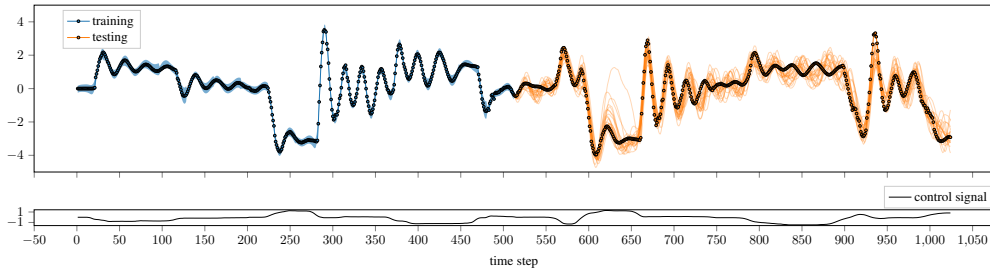

Figure 6: Demonstration of the identified model that controls the non-linear dynamics of the actuator dataset. The model's fitting on the train data and sampled future predictions, after applying the control signal to the system. Ground truth is denoted with the marked lines.

shows the fitting on the train data along with sampled future predictions from the learnt system when operating on a free simulation mode. It is worth noting the correct capturing of the uncertainty from the model at the points where the predictions are not accurate.

## 5    Discussion and conclusion

We have proposed a novel inference mechanism for the GPSSM, in order to address the challenging task of non-linear system identification. Since our inference is based on the variational framework, successful learning of the model relies on defining good approximations to the posterior of the latent functions and states. Approximating the posterior over the dynamics with a sparse GP seems to be a reasonable choice given our assumptions over the transition function. However, the difficulty remains in the selection of the approximate posterior of the latent states. This is the key component that enables successful learning of the GPSSM.

In this work, we construct the variational posterior so that it follows the same Markov properties as the true states. Furthermore, it is enforced to have a simple-to-learn, linear, time-varying structure. To assure, though, that this approximation has rich representational capacity we proposed to recover the variational parameters of the posterior via a non-linear recurrent recognition model. Consequently, the joint approximate posterior resembles the behaviour of the true system, which facilitates the effective learning of the GPSSM.

In the experimental section we have provided evidence that the proposed approach is able to identify latent dynamics in true and simulated data, even from partial and lagged observations, while requiring only small data sets for this challenging task.

**Acknowledgement**

Marc P. Deisenroth has been supported by a Google faculty research award.

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
