[Supplementary Material · supp_mat.pdf]

# A   Derivation of the ELBO

This appendix contains three parts: we first explicate the joint distribution of the model and data $p(\boldsymbol{X}, \boldsymbol{f}(\cdot), \boldsymbol{Y})$; then we describe the variational approximation to the model posterior $q(\boldsymbol{X}, \boldsymbol{f}(\cdot))$; then we show how they combine to produce the ELBO. Table 2 provides some nomenclature.

Table 2: Nomenclature used in this derivation

| | |
|---|---|
| $t \in \{0 \dots T\}$ | time steps indexed $t$ |
| $d \in \{1 \dots D\}$ | dimension of hidden states $\boldsymbol{x}_t$ indexed $d$ |
| $O$ | dimension of the observed data |
| $m \in \{1 \dots M\}$ | number of inducing variables indexed $m$ |
| $\boldsymbol{x}_t$ | hidden state at time $t$, $\boldsymbol{x}_t \in \mathbb{R}^D$ |
| $\boldsymbol{a}_t$ | control input (action) at time $t$, $\boldsymbol{a}_t \in \mathbb{R}^P$ |
| $\tilde{\boldsymbol{x}}_t$ | concatenation of control input and state at $t$ |
| $\boldsymbol{y}_t$ | observation at time $t$, $\boldsymbol{y}_t \in \mathbb{R}^O$ |
| $\tilde{\boldsymbol{y}}_t$ | concatenation of control input and observation at $t$ |
| $\boldsymbol{X}$ | collection of hidden states, $\boldsymbol{X} = [\boldsymbol{x}_t]_{t=0}^T$. |
| $\boldsymbol{Y}$ | collection of observations, $\boldsymbol{Y} = [\boldsymbol{y}_t]_{t=0}^T$. |
| $\sigma_f^2$ | variance of state transition noise |
| $\sigma_g^2$ | variance of observation nosie |
| $f_d(\cdot)$ | the $d^{\text{th}}$ Gaussian process (GP) |
| $\boldsymbol{f}(\cdot)$ | collection of GPs, $= [f_d(\cdot)]_{d=1}^D$ |
| $\eta_d(\cdot)$ | prior mean function of the $d^{\text{th}}$ GP |
| $k_d(\cdot, \cdot)$ | prior covariance function of the $d^{\text{th}}$ GP |
| $\mu_d(\cdot)$ | posterior mean function of the $d^{\text{th}}$ GP |
| $v_d(\cdot, \cdot)$ | posterior covariance function of the $d^{\text{th}}$ GP |
| $\boldsymbol{Z}$ | Locations of variational pseudo-inputs |
| $\boldsymbol{u}_d$ | evaluations of the $d^{\text{th}}$ GP at the pseudo-inputs: $\boldsymbol{u}_d = [f_d(\boldsymbol{z}_m)]_{m=1}^M$. |
| $\boldsymbol{U}$ | collection: $\boldsymbol{U} = [\boldsymbol{u}_d]_{d=1}^D$ |
| $\boldsymbol{\mu}_d$ | variational posterior mean of $\boldsymbol{u}_d$: $\boldsymbol{\mu}_d = [\mu_d(\boldsymbol{z}_m)]_{m=1}^M$ |
| $\boldsymbol{\Sigma}_d$ | variational posterior covariance of $\boldsymbol{u}_d$ |
| $\boldsymbol{A}_t$ | variational transition matrix of $q(\boldsymbol{x}_t \mid \boldsymbol{x}_{t-1})$ |
| $\boldsymbol{L}_t$ | triangular-square-root of variational covariance of $q(\boldsymbol{x}_t \mid \boldsymbol{x}_{t-1})$ |
| $\boldsymbol{m}_t$ | variational mean of the marginal $q(\boldsymbol{x}_t)$ |
| $\boldsymbol{S}_t$ | variational covariance of the marginal $q(\boldsymbol{x}_t)$ |

## A.1   Model joint distribution

Here we define the joint distribution of the Gaussian processes $f$, the latent states $\boldsymbol{x}$ and the data $\boldsymbol{y}$.

The Gaussian processes have prior mean $\eta(\cdot)$ and prior covariances $k(\cdot, \cdot)$:

$$p(f_d(\cdot)) = \mathcal{GP}\big(\eta_d(\cdot),\, k_d(\cdot, \cdot)\big) \quad d = 1 \dots D. \tag{17}$$

We note that placing a measure $p$ on the function $f$ causes some measure-theoretic discrepancies. Nonetheless, the derivation holds following a more theoretical consideration of the problem (Matthews et al., 2016), and the intuition given by our derivation is correct.

The initial state is assumed to be drawn from a standard normal distribution

$$p(\boldsymbol{x}_0) = \mathcal{N}(\boldsymbol{0},\, \boldsymbol{I}_D). \tag{18}$$

The state transition depends on the Gaussian processes:

$$p(\boldsymbol{x}_t \mid \boldsymbol{x}_{t-1}, \boldsymbol{f}(\cdot)) = \mathcal{N}\big(\boldsymbol{x}_t \mid \boldsymbol{f}(\tilde{\boldsymbol{x}}_{t-1}),\, \sigma_f^2 \boldsymbol{I}_D\big), \tag{19}$$

We assume a linear-Gaussian observation model:

$$p(\boldsymbol{y}_t \,|\, \boldsymbol{x}_t) = \mathcal{N}\big(\boldsymbol{y}_t \,|\, \boldsymbol{W}_g \boldsymbol{x}_t + \boldsymbol{b}_g, \sigma_g^2 \boldsymbol{I}_O\big) \tag{20}$$

The joint density is then

$$p(\boldsymbol{f}, \boldsymbol{X}, \boldsymbol{Y}) = \prod_{d=1}^{D} p(f_d(\cdot))\, p(\boldsymbol{x}_0) \prod_{t=1}^{T} p(\boldsymbol{y}_t \,|\, \boldsymbol{x}_t) \prod_{t=1}^{T} p(\boldsymbol{x}_t \,|\, \boldsymbol{f}, \boldsymbol{x}_{t-1}) \tag{21}$$

## A.2 Approximate posterior distribution

We will use variational Bayes to approximate the posterior distribution over $\boldsymbol{f}$ and $\boldsymbol{X}$, whilst simultaneously obtaining a bound on the marginal likelihood (the ELBO) which will be used to train the parameters of the model, including covariance function parameters, noise variances and the parameters $\boldsymbol{W}_g, \boldsymbol{b}_g$ of the linear output mapping.

The posterior over Gaussian processes takes the form of a sparse GP. We introduce a series of $M$ variational inducing points $\boldsymbol{Z} = [\boldsymbol{z}_m]_{m=1}^{M}$ which lie in the same domain as $\tilde{\boldsymbol{x}}$. Following convention, the values of the $d^{\text{th}}$ function at those points are denoted $\boldsymbol{u}_d = [f_d(\boldsymbol{z}_m)]_{m=1}^{M}$, while evaluations from the prior as $\boldsymbol{\eta}_d = [\eta_d(\boldsymbol{z}_m)]_{m=1}^{M}$. Note that the variables $\boldsymbol{u}$ are not *auxiliary* variables, but part of the original model specification, being part of the GP. We assume a variational posterior of the form

$$q(\boldsymbol{U}) = \prod_{d=1}^{D} \mathcal{N}\big(\boldsymbol{u}_d \,|\, \boldsymbol{\mu}_d, \boldsymbol{\Sigma}_d\big). \tag{22}$$

The remainder of the GPs conditioned on $\boldsymbol{u}$ are assumed to take the same form as the GP prior conditional. That is

$$q(f_d(\cdot) \,|\, \boldsymbol{u}_d) = p(f_d(\cdot) \,|\, \boldsymbol{u}_d) = \mathcal{GP}\big(\eta_d(\cdot) + k(\cdot, \boldsymbol{Z}) \boldsymbol{K}_{zz}^{-1}(\boldsymbol{u}_d - \boldsymbol{\eta}_d),\, k(\cdot, \cdot) - k(\cdot, \boldsymbol{Z}) \boldsymbol{K}_{zz}^{-1} k(\boldsymbol{Z}, \cdot)\big). \tag{23}$$

Marginalising with respect to $\boldsymbol{u}_d$ leads to our approximation to the GP:

$$q(f_d(\cdot)) = \mathcal{GP}\big(\mu_d(\cdot), v_d(\cdot, \cdot)\big), \tag{24}$$

with

$$\mu_d(\cdot) = \eta_d(\cdot) + k(\cdot, \boldsymbol{Z}) \boldsymbol{K}_{zz}^{-1}(\boldsymbol{\mu}_d - \boldsymbol{\eta}_d), \tag{25}$$

$$v_d(\cdot, \cdot) = k(\cdot, \cdot) - k(\cdot, \boldsymbol{Z}) \boldsymbol{K}_{zz}^{-1}[\boldsymbol{K}_{zz} - \boldsymbol{\Sigma}_d] \boldsymbol{K}_{zz}^{-1} k(\boldsymbol{Z}, \cdot). \tag{26}$$

The approximation to the posterior over state trajectories is given a Gauss-Markov structure of the form

$$q(\boldsymbol{X}) = q(\boldsymbol{x}_0) \prod_{t=1}^{T} q(\boldsymbol{x}_t \,|\, \boldsymbol{x}_{t-1}), \tag{27}$$

where

$$q(\boldsymbol{x}_0) = \mathcal{N}(\boldsymbol{x}_0 \,|\, \boldsymbol{m}_0,\, \boldsymbol{L}_0 \boldsymbol{L}_0^\top) \tag{28}$$

$$q(\boldsymbol{x}_t \,|\, \boldsymbol{x}_{t-1}) = \mathcal{N}(\boldsymbol{x}_t \,|\, \boldsymbol{A}_t \boldsymbol{x}_{t-1},\, \boldsymbol{L}_t \boldsymbol{L}_t^\top). \tag{29}$$

The complete set of variational parameters is then $\boldsymbol{Z}, \{\boldsymbol{\mu}_d, \boldsymbol{\Sigma}_d\}_{d=1}^{D}, \boldsymbol{m}_0, \boldsymbol{L}_0, \{\boldsymbol{A}_t, \boldsymbol{L}_t\}_{t=1}^{T}$. The parameters of $q(\boldsymbol{X})$ are reconfigured to be the output of an RNN recognition model (see main text), whilst we optimise the parameters controlling $\boldsymbol{f}(\cdot)$ directly.

The joint posterior then factors as

$$q(\boldsymbol{f}(\cdot), \boldsymbol{X}) = \prod_{d=1}^{D} q(f_d(\cdot)) q(\boldsymbol{X}). \tag{30}$$

## A.3 The ELBO

Having specified the forms of the model and the approximate posterior, we are ready to derive the ELBO. Following the standard variational Bayes methods, we write

$$\text{ELBO} = \mathbb{E}_{q(\boldsymbol{X})q(\boldsymbol{f}(\cdot))}\left[\log\frac{p(\boldsymbol{Y}\,|\,\boldsymbol{X})p(\boldsymbol{X}\,|\,\boldsymbol{f}(\cdot))}{q(\boldsymbol{X})}\frac{p(\boldsymbol{f}(\cdot))}{q(\boldsymbol{f}(\cdot))}\right]. \tag{31}$$

We will split the ELBO into four parts, dealing with each in turn:

$$\text{ELBO} = \underbrace{\mathbb{E}_{q(\boldsymbol{X})}\big[\log p(\boldsymbol{Y}\,|\,\boldsymbol{X})\big]}_{\text{part 1}} + \underbrace{\mathbb{E}_{q(\boldsymbol{X})q(\boldsymbol{f}(\cdot))}\big[\log p(\boldsymbol{X}\,|\,\boldsymbol{f}(\cdot))\big]}_{\text{part 2}}$$

$$-\underbrace{\mathbb{E}_{q(\boldsymbol{X})}\big[\log q(\boldsymbol{X})\big]}_{\text{part 3}} + \underbrace{\mathbb{E}_{q(\boldsymbol{f}(\cdot))}\big[\log\frac{p(\boldsymbol{f}(\cdot))}{q(\boldsymbol{f}(\cdot))}\big]}_{\text{part 4}}. \tag{32}$$

**Part 1** This expression can be computed straight-forwardly in closed form due to our choice of a linear-Gaussian emission $g(\boldsymbol{x})$. Let $\boldsymbol{m}_t, \boldsymbol{\Sigma}_t$ be the marginals of $q(\boldsymbol{x}_t)$ computed via the recursion, and recall the form of the linear emission function $g(\boldsymbol{x}_t) = \boldsymbol{W}_g\boldsymbol{x}_t + \boldsymbol{b}_g$

$$\mathbb{E}_{q(\boldsymbol{X})}\big[\log p(\boldsymbol{Y}\,|\,\boldsymbol{X})\big] = \mathbb{E}_{q(\boldsymbol{X})}\Big[\sum_{t=1}^{T}\log\mathcal{N}\big(\boldsymbol{y}_t\,|\,g(\boldsymbol{x}_t),\sigma_g^2\big)\Big]$$

$$= \sum_{t=1}^{T}\mathbb{E}_{q(\boldsymbol{x}_t)}\Big[\log\mathcal{N}\big(\boldsymbol{y}_t\,|\,\boldsymbol{W}_g\boldsymbol{x}_t + \boldsymbol{b}_g,\sigma_g^2\big)\Big]$$

$$= \sum_{t=1}^{T}\log\mathcal{N}\big(\boldsymbol{y}_t\,|\,\boldsymbol{W}_g\boldsymbol{m}_t + \boldsymbol{b}_g,\sigma_g^2\big) - \tfrac{1}{2\sigma_n^2}\text{tr}(\boldsymbol{W}_g^{\top}\boldsymbol{W}_g\boldsymbol{\Sigma}_t). \tag{33}$$

In practise we defer this simple computation to the `variational_expectations` functionality in GPflow (Matthews et al., 2017).

**Part 2** This expression cannot be computed in closed form without restriction to the RBF kernel as in (Frigola, 2015). We eliminate the integral with respect to $\boldsymbol{f}$ here, and then use the reparameterisation trick to estimate the integral with respect to $\boldsymbol{X}$ (see main text).

$$\text{part 2} = \mathbb{E}_{q(\boldsymbol{X})q(\boldsymbol{f}(\cdot))}\big[\log p(\boldsymbol{X}\,|\,\boldsymbol{f}(\cdot))\big]$$

$$= \mathbb{E}_{q(\boldsymbol{X})q(\boldsymbol{f}(\cdot))}\Big[\log p(\boldsymbol{x}_0)\prod_{t=1}^{T}\mathcal{N}\big(\boldsymbol{x}_t\,|\,\boldsymbol{f}(\tilde{\boldsymbol{x}}_{t-1}),\sigma_f^2\boldsymbol{I}_D\big)\Big]$$

$$= \mathbb{E}_{q(\boldsymbol{x}_0)}\big[\log p(\boldsymbol{x}_0)\big] + \mathbb{E}_{q(\boldsymbol{X})q(\boldsymbol{f}(\cdot))}\Big[\sum_{t=1}^{T}\sum_{d=1}^{D}\log\mathcal{N}\big(\boldsymbol{x}_t^{(d)}\,|\,f_d(\tilde{\boldsymbol{x}}_{t-1}),\sigma_f^2\big)\Big]$$

$$= \mathbb{E}_{q(\boldsymbol{x}_0)}\big[\log p(\boldsymbol{x}_0)\big] + \mathbb{E}_{q(\boldsymbol{X})}\Big[\sum_{t=1}^{T}\sum_{d=1}^{D}\log\mathcal{N}\big(\boldsymbol{x}_t^{(d)}\,|\,\mu_d(\tilde{\boldsymbol{x}}_{t-1}),\sigma_f^2\big) - \tfrac{1}{2}\sigma_f^{-2}v_d(\tilde{\boldsymbol{x}}_{t-1},\tilde{\boldsymbol{x}}_{t-1})\Big], \tag{34}$$

which matches the term in the main text.

**Part 3** This corresponds to the entropy of $q(\boldsymbol{X})$. It is straightforward to derive:

$$-\mathbb{E}_{q(\boldsymbol{X})}\big[\log q(\boldsymbol{X})\big] = \text{H}[q(\boldsymbol{X})] = \frac{(T+1)D}{2}\log(2\pi e) + \sum_{t=0}^{T}\log(\det(\boldsymbol{L}_t)). \tag{35}$$

**Part 4** This final part is the Kullback-Leibler divergence between the prior and (approximate) posterior GPs. We first note that it can be written as a sum across dimensions $d$, and then that each

GP $f_d(\cdot)$ can be factored into two parts: $p(f_d(\cdot) \,|\, \boldsymbol{u}_d)p(\boldsymbol{u}_d)$ and similarly for $q$. This results in

$$\mathbb{E}_{q(\boldsymbol{f}(\cdot))}\left[\log\frac{p(\boldsymbol{f}(\cdot))}{q(\boldsymbol{f}(\cdot))}\right] = \sum_{d=1}^{D}\mathbb{E}_{q(f_d(\cdot))}\left[\log\frac{p(f_d(\cdot))}{q(f_d(\cdot))}\right]$$

$$= \sum_{d=1}^{D}\mathbb{E}_{q(f_d(\cdot)\,|\,\boldsymbol{u}_d)q(\boldsymbol{u}_d)}\left[\log\frac{p(f_d(\cdot)\,|\,\boldsymbol{u}_d)p(\boldsymbol{u}_d)}{q(f_d(\cdot)\,|\,\boldsymbol{u}_d)q(\boldsymbol{u}_d)}\right] \;. \tag{36}$$

Since we have defined the posterior conditional $q(f_d(\cdot)\,|\,\boldsymbol{u}_d)$ to match the prior conditional, the two terms cancel, resulting in

$$\mathbb{E}_{q(\boldsymbol{f}(\cdot))}\left[\log\frac{p(\boldsymbol{f}(\cdot))}{q(\boldsymbol{f}(\cdot))}\right] = \sum_{d=1}^{D}\mathbb{E}_{q(\boldsymbol{u}_d)}\left[\log\frac{p(\boldsymbol{u}_d)}{q(\boldsymbol{u}_d)}\right]$$

$$= -\sum_{d=1}^{D}\mathrm{KL}\big[q(\boldsymbol{u}_d)||p(\boldsymbol{u}_d)\big] \;. \tag{37}$$

Since the result is a Kullback-Leibler divergence between two finite-dimensional normal distributions, it is computed straightforwardly.

Although this notation is somewhat sloppy (since the sets of variables $f_d(\cdot)$ and $\boldsymbol{u}_d$ overlap), the result is correct. Matthews (2017) contains a more careful and significantly more technical derivation.

# B Full visualisation of synthetic 1D dataset

Figure 7: Visualisation of the learned GP transition functions across a greater domain of the function. It can be seen that all models revert to the mean function (defined as the identity function) away from the data. The short lengthscales of the Arc-cosine and MGP (compounded with a Matern kernel) that are used to fit the kink of the true transition function mean that they almost instantaneously revert to the mean function. The longer length scales of the RBF-containing kernels mean that we revert much more slowly to the mean.

# C Learnt latent states for cart-pole

Below we provide the learnt latent states for the cart-pole dataset with observed and hidden velocities. It is worth noting that the model has recovered similar structure for both cases.

Figure 8: Learnt latent states for the cart-pole dataset with observed velocities.

Figure 9: Learnt latent states for the cart-pole dataset with hidden velocities.

# D Cart-pole training data fitting

Below we provide detailed fittings on the training episodes for the cart-pole dataset.

Figure 10: Detailed fittings per episode for the cart-pole dataset with observed velocities.

Figure 11: Detailed fittings per episode for the cart-pole dataset with hidden velocities.