[Reviews · NeurIPS 2017]

Reviewer 1



This paper is concerned with identification of Gaussian process state space models. The focus is on deriving a variational lower bound on the process combined with the reparameterization trick. The paper is generally well explain and the key concepts are demonstrated in the experiments. This kind of models have been extensively studied in the past. Similar constructions with linear measurement noise can be found in the literature, whereas the variational methodology to get a bound augmented with an RNN recognition model have been proposed by Frigola et al. and Mattos et al. respectively. The paper also follows a recent trend on building on past approaches by augmenting variational methods with the reparameterization trick. The above facts reduce the novelty of the approach. However, there are two key points to make here. Firstly, the reparameterization trick is particularly advantageous in this setting, because dynamical systems (in contrast to standard regression) very often need non-smooth and non-standard kernels. Secondly, although the model for the dynamics is linear (eq. 10,11), the recognition network parameters take up on driving the non-linear effects (eq. 12-14). The authors do not comment on that aspect, but I think it is a nice idea which can facilitate the optimization. The experiments disappointingly do not consider comparison with baselines and do not consider datasets from classical systems identification (e.g. Mattos et al. both for the method and the data). Other than this important omission, the results on the RL task are nice. == Post-rebuttal edit == I have read the authors' reply, thanks.

Reviewer 2



The paper proposes some advancements in the field of Gaussian process state space models. The main ideas are in the use of a recurrent neural network to parameterize the posterior over latent states. The paper is well written overall, although the use of a recurrent neural network model is not very well explained, and it assumes that the reader figures out some of the details. As a nonexpert in this particular domain and the proposed applications, I think that the idea is interesting, however I have some reservations about novelty. Most of the elements used to derive the inference scheme are based on standard variational inference, sparse GP approximations, and the reparameterization trick, and what is different is the parameterization of the posterior. Also, the proposed model follows from Frigola et al. (2014), so it feels as if this paper is a good realization of an incremental idea. The results are interesting but perhaps a more extensive evaluation would have been useful to strengthen the paper. The experiments are missing comparisons with simpler approaches (e.g., what if g() and/or f() are linear?) or any other competitor - without these, I believe is difficult to draw any conclusions on the potential impact of this work.

Reviewer 3



The authors derive a variational objective for inference and hyperparameter learning in a GPSSM. The authors apply a mean field variational approximation to the distribution over inducing points and a Gaussian approximation with Markov structure to the distribution over the sequence of latent states. The parameters of the latter depend on a bi-RNN. The variational bound is optimised using doubly stochastic gradient optimisation. The authors apply their algorithm to three simulated data examples, showing that particular applications may require the ability to flexibly choose kernel functions and that the algorithm recovers meaningful structure in the latent states. Overall, the paper is well written and provides an interesting combination of sparse variational GP approximations and methods for armotised inference in state space models. However, it is difficult to judge how much the proposed algorithm adds compared to existing sparse variational GPSSM models, since the authors did not make any comparisons or motivate the algorithm with an application for which existing approaches fail. From the paper alone, it is not clear whether related approaches could perform similarly on the given examples or what extensions of the model make it possible to go beyond what previous approaches can do. Comparisons to related models and a more quantitative analysis of the given algorithm would be essential for this. (The authors include RMSE comparisons to a Kalman filter and autoregressive GP in the rebuttal) Even though the motivation of the paper focuses on system identification, it would be interesting to see if the given approach is successful in correctly learning the model parameters (i.e. linear mapping of observations) and hyperparameters. Currently all results refer to the latent states or transition function. This would be interesting in order to evaluate biases that may be introduced via the choice of approximations and could also help with highlighting the need for the approximate inference scheme presented here (e.g. if a naive Gaussian parameterisation without the bi-RNN introduces more biases in learning). Lastly, the authors do not apply their algorithm to any real data examples, which again would be useful for motivating the proposed algorithm.